# Structural Modelling of KCNQ1 and KCNH2 Double Mutant Proteins, Identified in Two Severe Long QT Syndrome Cases, Reveals New Insights into Cardiac Channelopathies

**DOI:** 10.3390/ijms222312861

**Published:** 2021-11-28

**Authors:** William A. Agudelo, Sebastian Ramiro Gil-Quiñones, Alejandra Fonseca, Alvaro Arenas, Laura Castro, Diana Carolina Sierra-Díaz, Manuel A. Patarroyo, Paul Laissue, Carlos F. Suárez, Rodrigo Cabrera

**Affiliations:** 1Fundación Instituto de Inmunología de Colombia (FIDIC), Bogotá 111221, Colombia; wagudelos@gmail.com (W.A.A.); mapatarr.fidic@gmail.com (M.A.P.); cfsuarezm@gmail.com (C.F.S.); 2Laboratorio de Biología Molecular y Pruebas Diagnósticas de Alta Complejidad, Fundación Cardioinfantil-Instituto de Cardiología, Bogotá 111221, Colombia; sebastian.gil@urosario.edu.co (S.R.G.-Q.); a.fonseca2974@uniandes.edu.co (A.F.); 3Department of Pediatric Electrophysiology, Congenital Heart Institute, Fundación Cardioinfantil-Instituto de Cardiología, Bogotá 111221, Colombia; aarenas@lacardio.org (A.A.); lalaco2006@msn.com (L.C.); 4Center for Research in Genetics and Genomics (CIGGUR), GENIUROS Research Group, School of Medicine and Health Sciences, Universidad del Rosario, Bogotá 111221, Colombia; diana.sierra@urosario.edu.co (D.C.S.-D.); plaissue@biopasgroup.com (P.L.); 5Health Sciences Division, Main Campus, Universidad Santo Tomás, Bogotá 111221, Colombia; 6Microbiology Department, Faculty of Medicine, Universidad Nacional de Colombia, Bogotá 111221, Colombia; 7Orphan Diseases Unit, Biopas Laboratories, BIOPAS Group, Bogotá 111221, Colombia

**Keywords:** long QT syndrome, *KCNQ1*, *KCNH2*, cardiac channelopathies-computational modelling

## Abstract

Congenital long QT syndrome (LQTS) is a cardiac channelopathy characterized by a prolongation of the QT interval and T-wave abnormalities, caused, in most cases, by mutations in KCNQ1, KCNH2, and SCN5A. Although the predominant pattern of LQTS inheritance is autosomal dominant, compound heterozygous mutations in genes encoding potassium channels have been reported, often with early disease onset and more severe phenotypes. Since the molecular mechanisms underlying severe phenotypes in carriers of compound heterozygous mutations are unknown, it is possible that these compound mutations lead to synergistic or additive alterations to channel structure and function. In this study, all-atom molecular dynamic simulations of KCNQ1 and hERG channels were carried out, including wild-type and channels with compound mutations found in two patients with severe LQTS phenotypes and limited family history of the disease. Because channels can likely incorporate different subunit combinations from different alleles, there are multiple possible configurations of ion channels in LQTS patients. This analysis allowed us to establish the structural impact of different configurations of mutant channels in the activated/open state. Our data suggest that channels with these mutations show moderate changes in folding energy (in most cases of stabilizing character) and changes in channel mobility and volume, differentiating them from each other and from WT. This would indicate possible alterations in K^+^ ion flow. Hetero-tetrameric mutant channels showed intermediate structural and volume alterations vis-à-vis homo-tetrameric channels. These findings support the hypothesis that hetero-tetrameric channels in patients with compound heterozygous mutations do not necessarily lead to synergistic structural alterations.

## 1. Introduction

Congenital long QT syndrome (LQTS) is a cardiac channelopathy characterized by a prolongation of the QT interval and T-wave abnormalities on the electrocardiogram (ECG). It is commonly associated with syncope, seizures, and an increased risk of sudden cardiac death secondary to ventricular arrhythmias [1]. LQTS is caused by alterations in the genes encoding potassium, sodium, or calcium channels or adaptor proteins, which determine the duration of action potential [2]. Defects that lessen repolarizing K+ currents or enhance depolarizing Na+ and Ca++ currents can lengthen the action potential and produce a prolonged QT interval [3].

The most frequent molecular defects, which are present in 80–90% of genotyped cases, occur in *KCNQ1*, *KCNH2*, and *SCN5A* genes [4,5,6]. LQT1 is caused by loss-of-function mutations in *KCNQ1*, which encodes the α subunit of the slowly activating potassium channel Kv7.1, leading to reduced I_ks_ current. In physiological conditions, I_ks_ is increased by sympathetic stimulation and is essential for QT adaptation during tachycardia. LQT2 is associated with loss-of-function mutations in *KCNH2* (hERG) and a reduced rapidly activating potassium current I_kr_. LQT3 is caused by gain-of-function mutations in SCN5A resulting in increased I_Na_ current during the plateau and late phase of the action potential. Most defects associated with LQT1 and LQT2 are missense mutations, with <20% nonsense, frameshift, and splicing mutations, whereas the overwhelming majority of variants associated with LQT3 are missense and in-frame Indels [7].

Although the predominant pattern of LQTS inheritance is autosomal dominant (e.g., Romano-Ward syndrome), family studies enabling genetic mapping/genotyping can be challenging due to incomplete penetrance, epigenetic, and environmental factors [8]. Compound heterozygous mutations in genes encoding potassium channels have been reported in patients lacking family history, some of which are associated with early disease onset and more severe and treatment-refractory phenotypes (from 4.6% to 10%) [2,6,9,10]. This clinical presentation is often referred to as recessive Romano-Ward [11] or Jervell and Lange-Nielsen (JLN) syndromes, the latter defined by the presence of sensorineural hearing loss [12].

It is not clear why some mutations are able to cause disease in a heterozygous state whereas others are asymptomatic, and why some combinations of compound heterozygous mutations cause recessive Romano-Ward and others lead to Jervell and Lange-Nielsen (JLN) syndrome. This uncertainty makes genetic counselling of unaffected carriers challenging and can potentially lead to unnecessary interventions in healthy individuals. Genotype–phenotype correlation suggests that ion channels with certain single mutations retain sufficient residual function to render carriers unaffected unless there is an additional “hit” to the wild-type copy. Alternatively, it is possible that compound mutations interact in a synergistic manner to lead to much more severe alterations in compound heterozygotes than those observed in single mutants. Dissecting the contributions of different alleles to channel function can be challenging with traditional electrophysiological assays, whereas structural analyses can help determine the effect of distinct subunit interactions into channel function.

Protein modelling has been used to study the effect of different mutations and can be a tool for assessing risk in individual patients. Like other K_v_ channels, KCNQ1 and hERG have two general structural domains: Voltage-Sensing Domains (VSD) and a central Pore Domain (PD). The former responds to membrane depolarization and the latter contributes to ion selectivity. The central PD is surrounded by four VSDs, which give rise to an electromechanical coupling. A detailed structural understanding of KCNQ1 and hERG channels is needed to elucidate the molecular basis for channels’ loss-of-function in disease induced by point mutations [11]. However, it is not currently known how mutant channels encoded by compound heterozygote genotypes interact to produce abnormal complexes.

To better understand how compound heterozygous mutations can affect potassium channel function in severe cases of LQTS, all-atom molecular dynamic simulations were carried out, based on the structure for KCNQ1 channel, in activated/open state, reported by Kuenze et al. [13] and based on the structure for hERG reported by Wang et al. [14]. Similarly, we carried out these analyses on channels with different subunit combinations carrying mutations found in two cases of severe LQTS phenotypes with compound heterozygous mutations in potassium channels, with limited family history of arrhythmia or syncope. The structural analysis allowed us to explore channel mobility, channel pore radius/volume, and thermodynamic changes in protein stability (calculating mutation free energy cost).

## 2. Results

### 2.1. Sequencing and Variant Analysis

#### 2.1.1. Patient 1

Patient 1 carried two mutations in the *KCNQ1* gene and one on the *SCN5A*. The first variant on *KCNQ1* (p.Ala300Thr, c.898G>A) was unreported and of unknown significance and was also observed on the father; the second one (p.Ala344Val, c.1031C>T) has been reported as a disease-causing mutation [15] and has a de novo origin. Allele cloning revealed the p.Ala300Thr variant was of maternal origin and therefore present in a compound heterozygous configuration. The mutation found on *SCN5A* (p.Ala467Ser, c.1399G>T) was also observed on the mother and is of unknown significance (Figure 1 and Figure 2).

#### 2.1.2. Patient 2

Whole exome sequencing identified two variants in the *KCNH2* gene. The first one (p.Thr670Ile, c.2009C>T) was also observed in the mother. The other one (p.Pro507Thr, c.1519C>A) was also observed in the father. No variants were documented in the patient’s brother (Figure 1 and Figure 2). Both variants show a high level of sequence conservation and are predicted to be highly damaging by SIFT and Polyphen2.

### 2.2. Molecular Modelling

In general, the channel models remained stable in molecular dynamics and deviated by no more than a Cα RMSD of 4 to 10 Å from their respective starting conformations (see Appendix A). In the same way, the RMSF did not show big differences between WT and mutant channels even close to the mutation site. The mobility of regions where the mutations are, in both channels, was low in all RMSF profiles (Figure 3 and Figure 4). Regarding channel conformation, most remained in an open conformation (Figure 5), but small volume changes were seen that could affect the ion currents (Figure 6). The mutation free energy results show destabilization and stabilization effects (from −2 to 2 kcal/mol). This result, along with the above, could imply that folding and structuring are not the causes of channel malfunction.

#### 2.2.1. KCNQ1 Channel

The KCNQ1 channel had the p.Ala300Thr mutation near the ion selectivity filter, in the pore domain (P domain. Figure 3C). Based on the observations made by Butler et al. [16], when comparing wt and the p.Ser631Ala mutant based on cryo-EM hERG structure reported by Wang et al. [14], any small change in or near the ion selectivity filter can be decisive in the ion transit [16].

The other mutant, p.Ala344Val, was located in the S6 region, involved in the gating process, making the tetrameric form inactive [15]. The RMSF showed that this region is also a low mobility region.

The main difference in mobility among the models occurred in the S2-S3 loop (Figure 3C); no allosteric effects related with mutant regions were apparent, and this effect can be the consequence of the setup of PIP2 (phosphatidyl-4,5-bisphosphate) molecules at the inner membrane leaflet. This domain is a PIP2-interacting region [11].

Regarding radius/volume channel pore, KCNQ1 mutant channels tended to have a radius variation at 25Å *Z*-axis height (Figure 5A). This change had an effective impact on the channel’s volume. In this channel, the content of type-2 mutant (p.Ala344Val) has a proportional effect in the volume: the higher the content, the lower the volume (Figure 6A). All mutant combinations have statistically significant differences with WT (Figure 6C).

Regarding mutation free energy for the KCNQ1 channel, both mutants had stabilizing effects (positive ∆∆G). If these values are compared with studies done by Quan et al. [17], 2.56 kcal/mol can be considered a high stabilizing value (p.Ala300Thr) and could be due to the hydrogen bond capability of the threonine side chain compared with the alanine side chain. The 1.20 kcal/mol can be considered a medium-low stabilizing value, in line with no gain of hydrogen bonds and a change of amino acid volume (p.Ala344Val).

The p.Ala344Val mutant in KCNQ1 channel has been studied by Siebrands et al. [15]. They have shown that the mutant channel has a depolarizing shift of the voltage dependence of channel activation and an induction of voltage-dependent of channel inactivation, with respect to the wildtype. These changes are more pronounced when it interacts with KCNE1 accessory protein. This activation voltage shift affects the channel gating process via interactions between the S6 helix and S4-S5 linker domains. The channel volume decrease observed in our simulations is in accordance with the structural destabilization of the open state proposed in other works [15,18].

#### 2.2.2. hERG Channel

The hERG channel in this work had the p.Pro507Thr mutation in the S3-S4 loop of the VSD domain and the other mutation, p.Thr670Ile, was in the C-linker domain (Figure 4). Both are very far from the selectivity filter domain, so any affectation in the ion current can be expected as a product of non-trivial molecular interactions.

The mobility of loop S3-S4 is high in any channel (including WT) as well as the S1-S2 extracellular loop and the loop before P domain (around residue 600). In contrast, the S5 helix, S6 helix, and C-linker domains are less mobile (Figure 4C). Although there are no major changes in the mobility trends between the different models analyzed, there are significant differences in the pore radius, the WT having the smallest one (Figure 5B).

As can be seen in Figure 6C, the volume of the mutant channels is larger than that of the WT. This effect is most apparent in homo-tetrameric mutant channel 1111. They are also well differentiated in mobility patterns and the Cross Section Radii Curve (Figure 4C and Figure 5D, respectively), and any channel containing a subunit with mutant 1 (p.Pro507Thr) will have a larger volume than the homopolymer of mutant 2 (p.Thr670Ile) and WT. These differences are observed regardless of the Z-axis distance considered for their estimation (see Appendix A). The observation that the mutant channels have a higher volume should play a role in the gating process (activation/deactivation/inactivation), i.e., the ion current could be affected. Volume differences among channels are in the range of 1.45 times the wild type volume (medians of hERG-mt-1112/hERG-wt = 2586Å3/1781Å3).

The stability of channel mutants is clearly different (Table 1,). The VDS domain mutant p.Pro507Thr is destabilizing. Although there is a gained hydrogen bond, it is in a membrane accessible region, not solvent accessible. Elimination of the backbone blocking effect of proline appears to be the predominant factor of destabilization of loop S3-S4. However, the above is not reflected in changes in domain mobility (Figure 4C). The C-link domain mutant p.Thr670Ile is medium-low stabilizing. It is next to the C-terminal of S6 helix, a region of interaction with S4-S5 linker, key to the open-closed change of the channel [16].

The principal mechanism of loss of function in hERG by mutation is trafficking defects (protein trafficking or protein misfolding), leading to negative dominance or haploinsufficiency effects [11]. Given that the parents do not show noticeable phenotypic alterations, this could be ruled out in this case, indicating that the causes of the observed pathology could be related to changes involving the double mutant condition of studied cases. The changes in channel volume and stability implied by both mutant subunits lead to the inability to adequately reconstitute the function, whereas singly mutated channels present in the parents seem able to function normally.

## 3. Discussion

Electrophysiological characterization has previously been performed for both mutations present in KCNQ1 in patient 1. When expressed in Xenopus oocytes, the channel with the p.Ala300Thr mutation showed a significant hyperpolarizing shift of the activation voltage curve but also faster activation, which is likely to attenuate the reduction in outward K^+^ current at repolarizing potentials [19]. Similarly, the p.Ala344Val mutation is known to cause LQTS and expression in Xenopus oocytes displayed a voltage-dependent inactivation of the macroscopic current with no detectable alterations of maximal current amplitudes and increased sensitivity to local anesthetic inhibition [15]. These findings are consistent with evidence from our structural modelling, with a significantly reduced pore volume and radii observed in the homo-tetrameric channel for variant p.Ala344Val (MT-2222), likely accounting for voltage dependent inactivation. On the other hand, a near normal pore volume and radii in the homo-tetrameric channel for variant p.Ala300Thr (MT-1111) probably reflects a role in activation kinetics.

Electrophysiological changes induced by mutations found in KCNH2 in patient 2 have not been studied in vitro, but our modelling data suggest that there is an increase in pore volume and radii, mostly associated with channels containing subunits with the p.Thr670Ile variant. This suggests that reduced I_kr_ in this patient is most likely more affected by activation/inactivation dynamics than by pore permeability. However, one limitation of this study is that it is not yet able to study activation/inactivation dynamics in response to voltage, therefore limiting its ability to identify synergistic interactions between variants in these processes.

Both channels showed alterations in the pore and voltage sensing domains. This is similar to observations by Antúnez-Argüelles, where the same mutation observed here, in combination with p.Pro535Thr, was predicted to interfere with Calmodulin binding, as observed in a case of sudden death in an infant [20]. However, their analysis was unable to identify specific structural changes in hetero-tetrameric channels.

A double mutant can have a significant effect if the mutations occur in the same region of the protein [21]; however, any mutation, even a pair distant from each other, can modify the network of molecular interactions that determine the stability of a protein [22]. In this case, although each patient is a double mutant, no subunit of the channels has both mutations at the same time. In this context, given the severity of the clinical outcome, it is reasonable to assume a scenario in which the phenotype of the combination of the mutant subunits shows more than a simple additive effect. The simulation performed allowed us to evaluate additional hypotheses for the observed phenotype, such as additive, synergistic, or epistatic effects, allowing us to propose an additive rather than a synergistic effect.

Interestingly, in both cases, mutations did not seem to act synergistically in hetero-tetrameric channels to influence major alterations in protein mobility, or in pore volume/radius. This suggests that synergistic interactions between variants are not necessary to influence severe phenotypes and that a reduction in residual wild-type channel function is enough to go from an asymptomatic phenotype (in single mutant channels) to a severe phenotype (in compound heterozygous mutants).

Determining the effect of compound heterozygous mutations is difficult because the interaction between the protein defects are not completely understood, and in vitro assays cannot differentiate between the different possible configurations of tetrameric channels. Furthermore, because variants that do not cause disease in a heterozygous state are often misclassified as benign, they may cause disease as a part of a complex genotype [23]. In this study, it was observed that the phenotypes of hetero-tetrameric channels made up by different mutant alleles did not show synergistic structural alterations when compared with homo-tetrameric channels. This suggests that compound heterozygous mutations can lead to severe phenotypes without synergistic structural effects, presumably by abolishing residual wild-type function in hetero-tetrameric channels beyond a certain threshold.

Furthermore, since these genotypes confer a high risk of sudden death and are not readily identified through family studies because relatives are often asymptomatic, these complex genotypes are only identified when molecular autopsies are performed, which is rare [24]. However, this report and others suggest that they are important causes of the most severe phenotypes without family history of the disease [9]. We propose that this mode of inheritance may be relatively common in severe LQTS cases with limited or no family history of LQTS and should be considered when analyzing and counseling family members.

## 4. Materials and Methods

### 4.1. Subjects

#### 4.1.1. Patient 1

A 5-year-old male patient who consulted for episodes of syncope was referred by his mother, stating that for two years he has been presenting fainting, preceded by chest pain and skin pallor during exercise, associated with sphincter relaxation and hypotonia. The echocardiogram showed a structurally normal heart with adequate function. The ECG showed sinus bradycardia and a prolonged QTc of 610ms (QT interval corrected with Bazett’s formula). LQTS was diagnosed and β-Blocker treatment was initiated. The patient remained stable with β-Blockers in the subsequent controls.

Nevertheless, at 14 years old, the patient presented a new syncope episode despite treatment. Holter monitoring showed sinus rhythm, QTc of 590 ms, and bifid T waves. Due to the episodes of syncope, despite adequate pharmacological treatment, a dual-chamber cardiac defibrillator (DR364D Protecta, Medtronic Inc., Minneapolis, MN, USA) was implanted.

The patient did not present any episode of torsades de pointes (TdP) or ventricular fibrillation (VF) during his evaluation.

The family history was examined in detail through interviews with the patient’s parents and no heart conditions or history of LQTS, syncope, or sudden death were documented in any of the patient’s first- and second-degree family members. No significant consanguinity was reported in the family.

#### 4.1.2. Patient 2

A 9-year-old male who presented syncope during rest in decubitus was taken to the emergency room in cardiac arrest and ventricular fibrillation (VF) was documented. Cardiopulmonary resuscitation (CPR) was performed, reverting to sinus rhythm after 5 defibrillation shocks. The echocardiogram showed a structurally normal heart, with adequate function. The electrocardiogram (ECG) showed sinus rhythm with a prolonged QTc interval (QT corrected with Bazzet’s formula) of 588ms. LQTS was diagnosed and treatment with β-Blockers was initiated.

On the second day at hospital, he presented new episodes of cardiac arrhythmia, including torsades de pointes (TdP) and VF, that were resolved once again with CPR. In the next days despite the β-Blockers treatment, multiple events of VF were exhibited, hence an endocavitary bicameral cardiac defibrillator (Evera MRI, Medtronic Inc., Minneapolis, MN, USA) was implanted.

Maximum doses of β-Blockers were administered and persistent episodes of VF and TdP were observed, so thoracoscopic left cardiac sympathetic denervation (LCSD) was performed. No new ventricular arrhythmias episodes were displayed, and optimal cardiac defibrillator function was observed. Both parents showed normal QT intervals although the mother reported antecedents of convulsions and syncopes but with normal electrocardiographic studies. The patient’s brother did not present any cardiac alteration. The family history was examined in detail through interviews with the patient’s parents and no heart conditions or history of LQTS, syncope, or sudden death were documented in any of the patient’s first- and second-degree family members. No significant consanguinity was reported in the family.

### 4.2. Molecular Diagnosis

#### 4.2.1. Sequencing and Bioinformatics Analysis of Variants

As part of a cohort of LQTS patients [25], DNA extraction from the patients’ blood leukocytes was carried out with the QIAmp DNA Mini Kit (Qiagen, Hilden, Germany), according to the manufacturer’s instructions. Whole exome sequencing was carried out in collaboration with Macrogen Inc. (Seoul, Korea), as described in the Appendix A. Multiple alignments of 6 protein sequences were performed to assess the conservation during evolution of the altered residues.

#### 4.2.2. Homology Models

The mutations in positions p.Pro507Thr (variant 1) and p.Thr670Ile (variant 2) for the hERG channel were modelled using the 5VA1 structure [14] reported in the PDB database [26] as a starting point and models for mutations in positions p.Ala300Thr (variant 1) and p.Ala344Val (variant 2) in KCNQ1 were obtained from the work by Kuenze et al. (KCNQ1 Active-Open developed model) [13]. An important difference between both structural models is the absence of the C and N terminals in the KCNQ1 channel. The availability of templates for homology modeling of the KCNQ1 channel is limited, and modeling this region (considerably bulky but important for its interaction with PIP2 and calmodulin) [13] is beyond the scope of this study. On the other hand, the structure of the hERG channel is available by Cryo-EM [14], which facilitated the inclusion of this entire channel in the simulation. During the development of our work, Sun and MacKinnon reported work [27] with the complete structure of the KCNQ1 channel. We will take this structure into account for future work.

Substitutions were modeled using Drunback’s amino acid (aa) rotamer library using UCSF Chimera [28]. WT channels and channels built with all possible mutant subunit combinations were modelled for further analysis.

#### 4.2.3. Secondary Structure Motifs and SASA Analysis

The protein secondary structure motifs calculated with PDBsum [29,30] were computed using v.3.0 of Gail Hutchinson’s PROMOTIF program [31]. The Solvent Accessible Surface Areas (SASA) calculation was done with the GetArea tool from http://curie.utmb.edu/getarea.html [32] (accessed on 15 January 2020). A consensus solvent accessibility for each chain in the tetrameric channels was evaluated and the SASA for the primary sequence was summarized as shown in Figure 3 and Figure 4.

### 4.3. Molecular Dynamics

All-atom molecular dynamics simulations were made for each channel (14 total), with the aim of studying the effect of mutation in the inter domain mobility. The molecular systems simulated for channel hERG were three homo-tetrameric (hERG-wt, hERG-mt-1111 and hERG-mt-2222) and four hetero-tetrameric channels (hERG-mt-1112, hERG-mt-1122, hERG-mt-1212, and hERG-mt-1222). For channel KCNQ1, three homo-tetrameric (KCNQ1-wt, KCNQ1-mt-1111, and KCNQ1-mt-2222) and four hetero-tetrameric channels (KCNQ1-mt-1112, KCNQ1-mt-1122, KCNQ1-mt-1212, and KCNQ1-mt-1222) were simulated. These systems were selected according to the spectrum of phenotypes of the channels of the doubly mutant individuals. WT phenotypes were analyzed as a point of comparison for the analysis of mutant channels.

Each channel was placed in an explicit phospholipid membrane through the PPM method based on the OPM database [33]. For hERG channels, a bilayer of POPC (palmitoyloleoyl-phosphatidylcholine) (~270 lipids per leaflet) and for KCNQ1 channels a bilayer of POC and PIP2 (phosphatidyl-4,5-bisphosphate; 26 molecules at inner leaflet) was set up using the builder tool of the CHARMM-GUI website [34,35]. Although it is known that the main chains of these channels work together with other accessory proteins (KCNQ1/KCNE1 [36] and hERG/KCNE2 [11]), it is also known that the channels formed from the main chain maintain their function of transmitting current [11]. Due to the latter, and for the sake of simplicity, it was decided to work only with the main chain.

The TIP3P explicit water model was used to simulate solvent interactions [37], consisting of a layer with 22.5 Å thickness and 0,15 M of KCl, on either side of the membrane. The SHAKE algorithm was used to restrain the protein hydrogen atom mobility [38]. To provide an idea of the number of atoms, the hERG-wt system has 68,000 water molecules, 17,000 hydrogen atoms, and 12,000 heavy atoms (C, N, O, S). Amber14 software was used [39] with ff14SB [40] and Lipid 11 [41] force fields. The PIP2 lipid parameters were obtained from the work by Kuenze et al. [13].

For each channel, four independent replicas of molecular dynamics were done. Minimization 2500 steepest descent followed by 2500 conjugate gradient with restraints on protein atoms (10 kcal/Å^2^) and lipid head group (Phosphorus atom 2.5 kcal/Å^2^).

The equilibration phase consisted of 375 ps molecular dynamics at 300 K to release atomic constraints and allow solvent molecules to get into protein cavities. Finally, 100 ns molecular dynamics (0.002 ps time step) for data production were done. Data recollection was done each 10 ps. For molecular dynamics convergence calculation of RMSD with Cα coordinates was done. The RMSF analysis (residue mobility) was also performed with Cα coordinates.

### 4.4. Channel Pore Radius and Volume

The channel pore radius and pore volume were calculated with the aim of estimating any structural change that can affect the ion traffic and channel gating. For this purpose, molecular dynamics snapshots each 0.5 ns were taken for all channels and the pore radius was calculated using HOLE software [30]. The volume was calculated integrating the pore radius curve at various heights of the *Z*-axis. A 0–40 Å height range at *Z*-axis was selected for the summary analysis, since the main effects in the pore radius lie in this zone, previous to K^+^ selectivity filter.

### 4.5. Mutation Free Energy

A computational estimation of ΔΔG of mutation was carried out [17,42,43] in order to establish the change in stability of proteins due to amino acid mutation, i.e., how the mutant channels differ or not from wild type interactions; for example, loss or gain of hydrogen bond with other parts of the protein or with the solvent.

Technically, the change in stability of proteins is understood through a two-state folding model, where the change in the free energy of folding is assimilated as a change in its structure stability, due to the mutation. This change is calculated as ΔΔG = ΔG_mt_^folding^ − ΔG_wt_^folding^, and negative values implies that mutation is destabilizing; otherwise, it induces stabilization. Computationally, the ΔΔG can be estimated as shown in Figure 7.

For this simulation, the Thermodynamic Integration (TI) protocol implemented in Amber14 [44] was used. The protocol implies a molecular dynamics simulation over a mixed system generated from the wildtype system (wt) and the mutant system (mt). The mix grade is modulated by a parameter known as λ, that changes between 0 and 1 values: λ = 0 is the Wt system and λ = 1 is the Mt system; any other value implies a mixed system Wt-Mt. There are different mathematical forms for calculating mixed energy; a linear form Wt-Mt = (1 − λ) Wt + λ Mt was used here. For the sake of simplicity, only 3 values of λ:0, 0.5, 1 were analyzed. For each λ value it was necessary to run an independent molecular dynamics simulation, each simulation producing a specific dG/dλi average value. The Free Energy change of the process (in this case, mutation of one residue) was calculated by:(1)$$\Delta G_{\mathrm{wt}\to \mathrm{mt}}=\int_{0}^{1}\;\;\;\langle \frac{\mathrm{dG}}{d\lambda }\rangle d\lambda$$

A simple trapezoid rule for integration and for error propagation estimation was used.

## Figures and Tables

**Figure 1 ijms-22-12861-f001:**
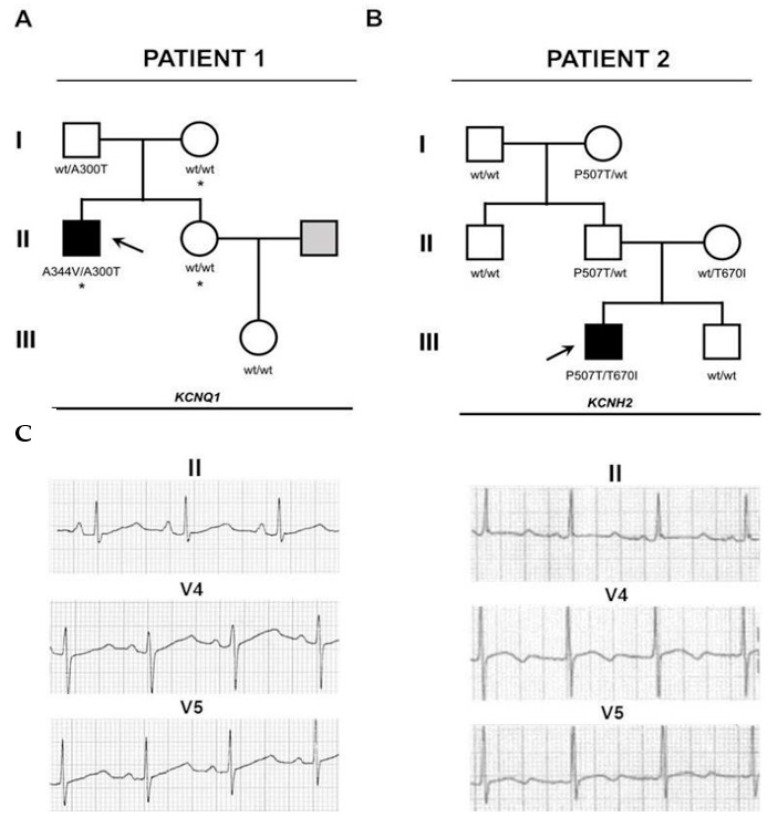
Family history and electrophysiological profile of the patients described in this study. (**A**,**B**) Familiograms showing the single affected patient in each family (arrow). Genotypes of tested individuals are shown. Individuals shaded white showed normal QT intervals. The individual shaded gray was not evaluated. The asterisk denotes patients with the *SCN5A* p.Ala467Ser, c.1399G>T variant. (**C**) Selected ECG derivations of the patients showing prolonged QTc interval, in the right panel, the QTc is 573 ms and the T wave is biphasic. In the left panel, the QTc is 524 ms and the T wave is bifid.

**Figure 2 ijms-22-12861-f002:**
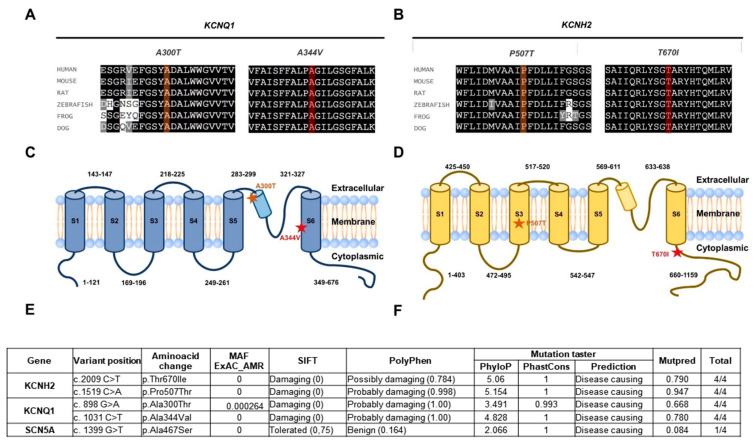
Sequence conservation, location, and predicted consequence of variants identified in the patients. Patient 1 (**A**,**C**) and Patient 2 (**B**,**D**). The predicted aminoacids (**A**,**B**) and the location (**C**,**D**, stars) of each variant are shown in orange and red. Conservative substitutions are shown in grey and non-conservative substitutions are shown in white. Location of variant residues from patients 1 and 2, respectively, in protein schematics depicting secondary structure. (**E**,**F**) Results of pathogenicity prediction analyses from SIFT, PolyPhen Mutation Taster, and Mutpred for all variants identified in these patients.

**Figure 3 ijms-22-12861-f003:**
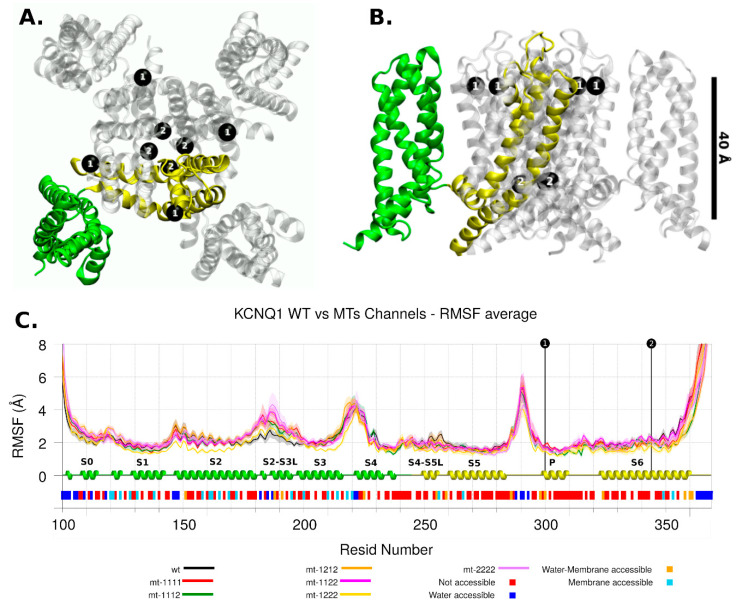
KCNQ1 Channel. The mutation location is shown with black dots and number 1 or 2. Two protein intermembrane domains are shown in green and yellow. (**A**) Top view. (**B**) Lateral view. (**C**) RMSF analysis for wild and mutant types. Each series corresponds to 4 chains and 4 replicas for each one, then it shows its average and standard deviation. The secondary structure motifs and solvent accessibility are shown below.

**Figure 4 ijms-22-12861-f004:**
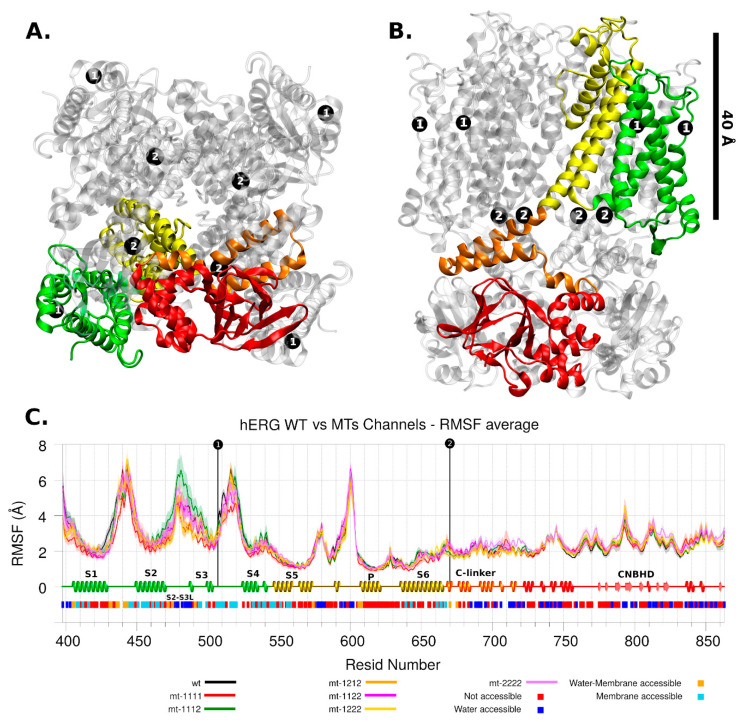
hERG Channel. The mutation location is shown with black dots and number 1 or 2. Four protein domains are shown in colors: intermembrane (green and yellow), inner side of the membrane (orange and red). (**A**) Top view. (**B**) Lateral view. (**C**) RMSF analysis for wild and mutant types. Each series corresponds to 4 chains and 4 replicas for each one, then it shows its average and standard deviation. The secondary structure motifs and solvent accessibility are shown below.

**Figure 5 ijms-22-12861-f005:**
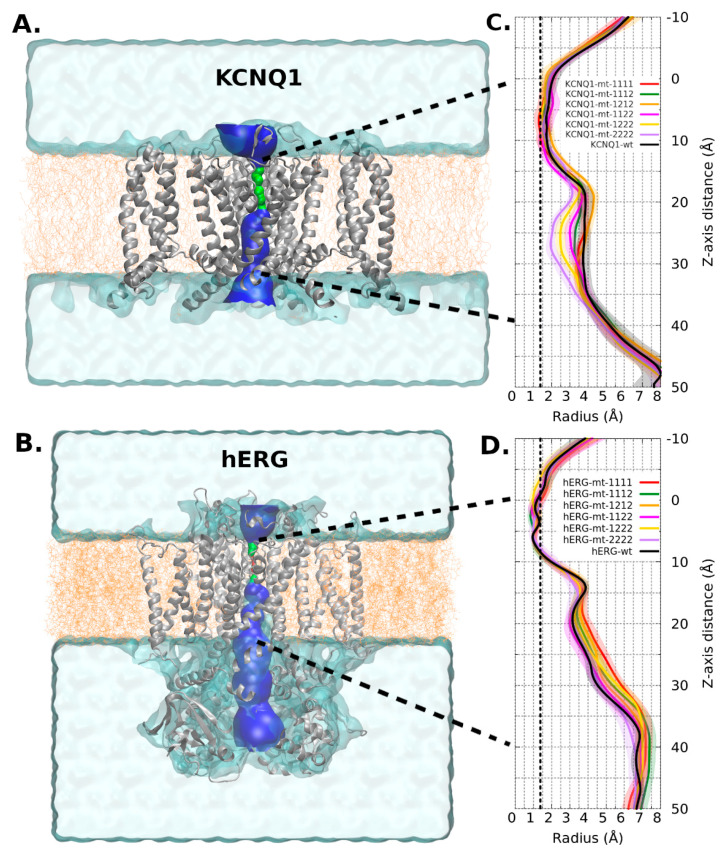
Membrane location, pore location, and cross section radii curve. (**A**,**B**): Membrane (orange) and solvent location (cyan surface region). Pore region: green surface shows region with one water molecule access and blue surface shows region with >2 water molecule access. (**C**,**D**): Cross section radii curves. Each series corresponds to 4 replicas, then it shows its average and standard deviation (as shaded area). The positive values of the Z-axis correspond to the inner side of the membrane. The vertical dotted line at 1.38 Å indicates the ionic radius of a K^+^ ion. Cross section raddi curves, corresponding to the limits depicted by bold dotted lines in A and B. The transmembrane pore comprises 0 to 40 Å.

**Figure 6 ijms-22-12861-f006:**
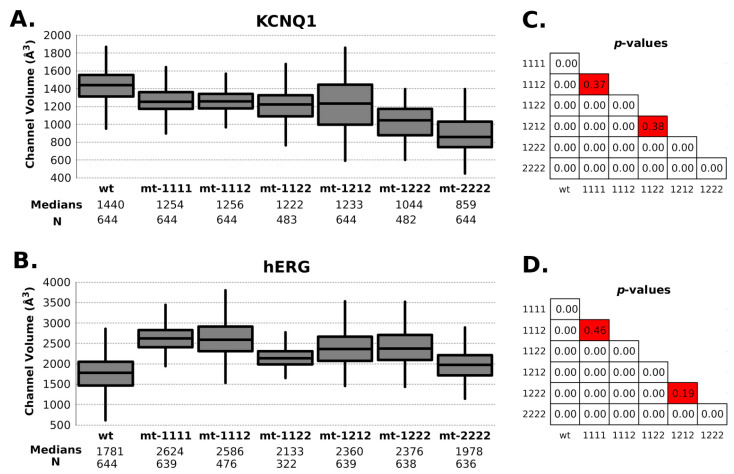
Channel volume. Box plots show the distribution of channel volume values calculated from molecular dynamics (**A**) KCNQ1 channel and (**B**). hERG channel. Panels (**C**). (KCNQ1) and (**D**). (hERG) show the Welsh t-test p-values between each data series. Boxes in red show nonsignificant differences between distributions. The time interval between each volume data is 0.5 nanoseconds.

**Figure 7 ijms-22-12861-f007:**
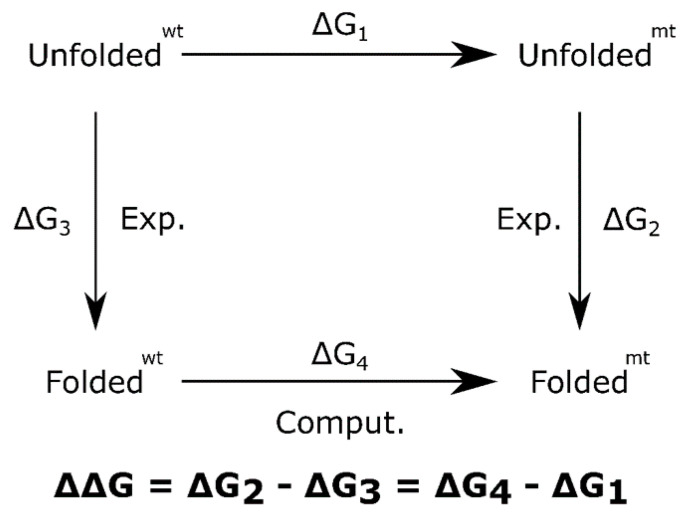
Thermodynamic cycle to calculate mutation free energy.

**Table 1 ijms-22-12861-t001:** Mutation free energy (monomer and tetramer). Values in parentheses are mean standard errors. KCNQ1: Mutant 1 = p.Ala300Thr; Mutant 2 = p.Ala344Val. hERG: Mutant 1 = p.Pro507Thr; Mutant 2 = p.Thr670Ile.

		∆∆G (kcal/mol)	
Channel	Mutant	Monomer	Tetramer
KCNQ1	mt-1111	2.56 (0.04)	10.24 (0.17)
	mt-2222	1.20 (0.04)	4.81 (0.17)
hERG	mt-1111	−2.71 (0.05)	−10.83 (0.21)
	mt-2222	1.14 (0.05)	4.58 (0.19)

## Data Availability

The data presented in this study are available on request from the corresponding author. The data are not publicly available due to the large amount of digital size (>500 GB). However, if there is any specific configuration of the channel that you want to evaluate, it can be extracted from the data set and requested from the authors.

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
