# Peer review of "Structural Modelling of KCNQ1 and KCNH2 Double Mutant Proteins, Identified in Two Severe Long QT Syndrome Cases, Reveals New Insights into Cardiac Channelopathies"

_ijms, 2021, doi:10.3390/ijms222312861_

Round 1

Reviewer 1 Report

Dear Editor,

I read with interest the paper from Agudelo et al. The authors provided a good case of novel mutations able to indentify patients with long QT syndrome and, contextually, provide new insights in the overall comprehension of the complex mechanisms at the base of channelopaties. The paper is really well written, the language is fluent as well as its scientidic background. The figures and tables are easy to be understood.

Author Response

Dear Mr. Kobe Guo:

We would like to thank the referees for taking the time to read our paper and make helpful suggestions for improving it. We are thus submitting a corrected version of our manuscript; in this document, we reply to the comments by reviewer 1. We hope that you find this improved version of the manuscript suitable for publication in your prestigious journal.

Best Regards

Reviewer’s comments:

I read with interest the paper from Agudelo et al. The authors provided a good case of novel mutations able to identify patients with long QT syndrome and, contextually, provide new insights in the overall comprehension of the complex mechanisms at the base of channelopathies. The paper is really well written, the language is fluent as well as its scientific background. The figures and tables are easy to be understood.

We thank the reviewer for the thoughtful analysis of our work and hope the revised manuscript addresses any concerns.

Reviewer 2 Report

The authors here used molecular simulations of two voltage-gated potassium channels (KCNQ1 and hERG) to investigate the structural impacts of different configurations of LQTS-associated mutant channels. They found that these mutations change the folding energy, channel mobility and volumes. They concluded that hetero-tetrameric channels with compound heterozygous mutations do not interact in a synergistic manner and lead to severe phenotypes. The data are convincing and the conclusions seem appropriate. But the manuscript is not well-presented. There are some concerns the authors might need to address.  

  1. Grammar errors below, which makes the paper hard to read

Abstract: since the molecular mechanisms underlying….

Method: Patient 1 carried two mutations in the KCNQ1….

Figure 4 legend:  Down is shown the secondary structure…

Page 8: being the tetrameric form inactive: what does this mean?

Page 8: They have shown that the mutant channel has a depolarizing shift of the voltage dependence…

All mutant combinations showed significant differences with WT.

LOF (loss-of-function)

  1. Where is figure 1?
  2. Can the authors explain the relation between KCNQ1 and IKs (KCNQ1/KCNE1), also for the hERG and IKr in the intro?
  3. The gene names should be italicized.
  4. Please label figure 2C in the Figure 2.
  5. Please explain why in Figure 4 the cytosolic domains of KCNQ1 were not analyzed, while in Figure 5, they were analyzed in hERG? In other words, why the C-linker and CNBHD in KCNQ1 were not studied?
  6. The figure legends need to be consistent with the text, for example, Figure 7B, 7C
  7. Some recent publications and reviews about the KCNQ1 structure, the hERG channel, KCNQ1 modulation by KCNE1 should be mentioned or acknowledged in the intro or discussion. For example,

https://doi.org/10.1016/j.cell.2019.12.003, https://doi.org/10.3390/ijms21249440,

  1. Both the double mutants in KCNQ1 and hERG channels, respectively, seem far from each other according to the structures, how could the authors assume they could have synergistic interactions in terms of the phenotypes?

Author Response

Dear Mr. Kobe Guo:

We would like to thank the referees for taking the time to read our paper and make helpful suggestions for improving it. We are thus submitting a corrected version of our manuscript; we have made point-by-point replies to all the comments by reviewer 2 and highlighted the changes made throughout this version of the manuscript. We hope that you find this improved version of the manuscript suitable for publication in your prestigious journal.

Best Regards

  1. Grammar errors below, which makes the paper hard to read

Response: Document was updated taking into account all reviewer inquiries:

  1. Abstract: since the molecular mechanisms underlying….

We have clarified this sentence; the change has been highlighted:

...Since the molecular mechanisms underlying severe phenotypes in carriers of compound heterozygous mutations are unknown,..

  1. Method: Patient 1 carried two mutations in the KCNQ1….

We have clarified this sentence; the change has been highlighted:

...Patient 1 carried two mutations in the KCNQ1 gene and one on the SCN5A.

  1. Figure 4 & 5 legend:  Down is shown the secondary structure…

We have clarified this sentence; the change has been highlighted:

...The secondary structure motifs and solvent accessibility are shown below.

  1. Page 8: being the tetrameric form inactive: what does this mean?

We have clarified this sentence; the change has been highlighted:

The other mutant, p.Ala344Val, was located in the S6 region, involved in the gating process, making the tetrameric form inactive [15].

  1. Page 8: They have shown that the mutant channel has a depolarizing shift of the voltage dependence…

            We have clarified this sentence; the change has been highlighted:

They have shown that the mutant channel has a depolarizing shift of the voltage dependence of channel activation and an induction of voltage-dependent of channel inactivation, with respect to the wildtype. These changes are more pronounced when it interacts with KCNE1 accessory protein.

  1. All mutant combinations showed significant differences with WT.

We have clarified this sentence; the change has been highlighted:

All mutant combinations have statistically significant differences with respect to WT (Figure 6C).

  1. LOF (loss-of-function)

            We have removed the acronym:

A detailed structural understanding of the KCNQ1 and hERG channels is needed to elucidate the molecular basis for the loss of function of the channels in diseases induced by point mutations.

  1. Where is figure 1?

The figures were renumbered. Figure 1 became Figure 7 and now it is in the final section 4.5 Mutation Free Energy.

  1. Can the authors explain the relation between KCNQ1 and IKs (KCNQ1/KCNE1), also for the hERG and IKr in the intro?

Response:  We have rewritten this section, in response to the reviewer's concern:

Congenital Long QT syndrome (LQTS) is a cardiac channelopathy characterized by a prolongation of the QT interval and T-wave abnormalities on the electrocardiogram (ECG). It is commonly associated with syncope, seizures and an increased risk of sudden cardiac death secondary to ventricular arrhythmias [1]. LQTS is caused by alterations in the genes encoding potassium, sodium or calcium channels or adaptor proteins, which determine the duration of action potential [3]. Defects that lessen repolarizing K+ currents or enhance depolarizing Na+ and Ca++ currents can lengthen the action potential and produce a prolonged QT interval [2].

The most frequent molecular defects, which are present in 80-90% of genotyped cases, occur in KCNQ1, KCNH2 and SCN5A genes [4]–[6]. LQT1 is caused by loss‐of‐function mutations in KCNQ1, which encodes the α subunit of the slowly activating potassium channel Kv7.1, leading to reduced Iks current. In physiological conditions, Iks is increased by sympathetic stimulation and is essential for QT adaptation during tachycardia. LQT2 is associated with loss‐of‐function mutations in KCNH2 (hERG) and a reduced rapidly activating potassium current Ikr. LQT3 is caused by gain‐of‐function mutations in SCN5A resulting in increased INa current during the plateau and late phase of the action potential. Most defects associated with LQT1 and LQT2 are missense mutations, with <20% nonsense, frameshift and splicing mutations, whereas the overwhelming majority of variants associated with LQT3 are missense and in-frame Indels [7].

  1. The gene names should be italicized.

Response: We have made the suggested correction; the changes have been highlighted.

  1. Please label figure 2C in the Figure 2.

Response:  We have corrected the label; the changes have been highlighted:

Sequence conservation, location and predicted consequence of variants identified in the patients. Patient 1 (A and C) and Patient 2 (B and D).

  1. Please explain why in Figure 4 the cytosolic domains of KCNQ1 were not analyzed, while in Figure 5, they were analyzed in hERG? In other words, why the C-linker and CNBHD in KCNQ1 were not studied?

Response:  An explanation has been included in the corresponding section

Homology models

The mutations in positions p.Pro507Thr (variant 1) and p.Thr670Ile (variant 2) for the hERG channel were modelled using the 5VA1 structure [14] reported in the PDB database [24] as a starting point and models for mutations in positions p.Ala300Thr (variant 1) and p.Ala344Val (variant 2) in KCNQ1 were obtained from the work by Kuenze et al. (KCNQ1 Active-Open developed model) [13]. An important difference between both structural models is the absence of the C and N terminals in the KCNQ1 channel. The availability of templates for homology modeling of the KCNQ1 channel is limited, and modeling this region, (considerably bulky but important for its interaction with PIP2 and calmodulin) [13] is beyond the scope of this study. On the other hand, the structure of the hERG channel is available by Cryo-EM [14], which facilitated the inclusion of this entire channel in the simulation.

  1. The figure legends need to be consistent with the text, for example, Figure 7B, 7C (now figure 6)

All mutant combinations have statistically significant differences with respect to WT (Figure 6C).

Figure 6. Channel Volume. Box plots show the distribution of channel volume values calculated from molecular dynamics (A. KCNQ1 channel, and B. hERG channel). Panels C. (KCNQ1) and D. (hERG) show the Welsh t-test p-values between each data series. Values shaded in red indicate nonsignificant differences between distributions. The time interval between each volume data is 0.5 nanosecond.

  1. Some recent publications and reviews about the KCNQ1 structure, the hERG channel, KCNQ1 modulation by KCNE1 should be mentioned or acknowledged in the intro or discussion. For example, https://doi.org/10.1016/j.cell.2019.12.003, https://doi.org/10.3390/ijms21249440,

Response: We have added the following lines in the sections 4.2.2 and 4.3

With respect to Sun et al 2020,  we added the following line:

During the development of our work, Sun and MacKinnon reported work [Sun et. al. 2019] with the complete structure of the KCNQ1 channel. We will take this structure into account for future work.

And with respect to Wu et al 2021, we added the following line:

Although it is known that the main chains of these channels work together with other accessory proteins (KCNQ1/KCNE1 [Wu et. el. 2020] and hERG/KCNE2 [11]), it is also known that the channels formed from the main chain maintain their function of transmitting current [11]. Due to the latter, and in the sake of simplicity, it was decided to work only with the main chain.

  1. Both the double mutants in KCNQ1 and hERG channels, respectively, seem far from each other according to the structures, how could the authors assume they could have synergistic interactions in terms of the phenotypes?

Response: We have added a paragraph to clarify this issue in the discussion.

A double mutant can have a significant effect if the mutations occur in the same region of the protein (Perez-Perez et al 2009), however, any mutation, even a pair distant from each other, can modify the network of molecular interactions that determine the stability of a protein (Kastritis et al. 2014). In this case, although each patient is a double mutant, no subunit of the channels has both mutations at the same time. In this context, given the severity of the clinical outcome, it is reasonable to assume a scenario in which the phenotype of the combination of the mutant subunits shows more than a simple additive effect. The simulation performed allowed us to evaluate additional hypotheses for the observed phenotype, such as additive, synergistic or epistatic effects, allowing us to propose an additive rather than a synergistic effect.

Round 2

Reviewer 2 Report

The authors address all my concerns.